# Angiogenesis and Lymphangiogenesis in Medulloblastoma Development

**DOI:** 10.3390/biology12071028

**Published:** 2023-07-21

**Authors:** Manon Penco-Campillo, Gilles Pages, Sonia Martial

**Affiliations:** Institute for Research on Cancer and Aging (IRCAN), Université Côte d’Azur, CNRS UMR 7284 and INSERM U1081, 33 Avenue de Valombrose, 06107 Nice, France; manon.penco-campillo@univ-cotedazur.fr (M.P.-C.); gilles.pages@univ-cotedazur.fr (G.P.)

**Keywords:** medulloblastoma, childhood, angiogenesis, lymphangiogenesis, metastasis

## Abstract

**Simple Summary:**

The therapeutic strategy against medulloblastoma (MB) is based on accurate diagnosis of the pathology, monitoring after surgery, treatment of advanced forms and adaptation of treatment in the event of recurrence. The aim is to propose effective therapies at diagnosis, to adapt first-line treatments according to the severity of the disease and to offer equally effective treatments in the event of a relapse. MB current therapeutic regimens are heavy and lead to severe disabilities (paralysis, speech impairment, etc.). Identifying the severity of MB at the time of diagnosis can reduce the intensity of treatment and limit its disabling effects. MB recurrences (about 30% of patients) result in MB metastasis and are usually fatal. Blood vessels, as well as lymphatic vessels seem to be key players in MB progression and metastasis. Knowing the parameters and the molecular agents responsible for this pejorative evolution could lessen or even eliminate the fatal problems.

**Abstract:**

Medulloblastoma (MB) is the most prevalent brain tumor in children. Although the current cure rate stands at approximately 70%, the existing treatments that involve a combination of radio- and chemotherapy are highly detrimental to the patients’ quality of life. These aggressive therapies often result in a significant reduction in the overall well-being of the patients. Moreover, the most aggressive forms of MB frequently relapse, leading to a fatal outcome in a majority of cases. However, MB is highly vascularized, and both angiogenesis and lymphangiogenesis are believed to play crucial roles in tumor development and spread. In this context, our objective is to provide a comprehensive overview of the current research progress in elucidating the functions of these two pathways.

## 1. Introduction

Medulloblastoma (MB) is a malignant embryonic tumor that develops in the cerebellum. Although it is a relatively rare cancer (500 new cases are diagnosed each year in the United States, and approximately 100 in France), MB is the most frequent and aggressive intracranial malignant pediatric tumor, accounting for approximately 25% of CNS tumors in children [1]. In contrast, MB is much less common in adults [2]. The median age of patients ranges between 5 and 7 years old, with a higher incidence in males (boy/girl ratio of 1.8/1).

MB is not a single disease but rather encompasses a diverse range of pathologies with significant heterogeneity. Initially, the severity of these variations was assessed based on histological criteria. However, with recent advances in sequencing and molecular genetics, our understanding of MB greatly improved. These recent data enabled an update of the World Health Organization (WHO) classification in 2016 defining four subgroups, each with unique genetic alterations, epigenetic modifications, transcription profiles and clinical characteristics: Wingless (WNT), Sonic Hedgehog (SHH), non-WNT/non-SHH (Group 3 and Group 4) [3]. In this classification, SHH tumors were further stratified based on the *TP53* gene (wild-type or mutant) status. This mutational status has a significant impact on prognosis and is correlated with distinct clinicopathologic characteristics. This classification partly aligns with the previous histopathological classification: WNT tumors are predominantly characterized by a classic morphology while desmoplastic/nodular MB and MB with extensive nodularity (MBEN) correspond to the SHH Group. Anaplastic large cell tumors, which often exhibit MYC amplification, are primarily classified under Group 3, with a few cases falling into a molecular subtype of the SHH Group [4].

The WNT and SHH subgroups are characterized by aberrant activation of the WNT and SHH signaling pathways, respectively, which play crucial roles in the pathogenesis of these groups. However, no specific signaling pathway appears to play a similar tumorigenic role in the two other groups. Both Groups 3 and 4 exhibit distinct molecular characteristics including the overexpression of N-myc and c-myc factors and the inactivation of *TP53* [5].

Meta-analyses showed clear distinctions among these four subgroups in terms of histology, chromosomal aberrations, and clinical prognosis [6]. Prognosis prediction is more reliable than with the previous histopathological classification. 

The current consensus officially recognizes four MB subgroups, although biological heterogeneity exists both within and between subgroups [7]. An integrated genomic analysis of 194 primary tumors (validated on three independent cohorts) revealed the presence of highly aggressive intermediate tumors, which belong to specific “subsets” of Group 3 or 4 [8]. However, these subsets are not yet well characterized.

In 2017, three independent studies identified several molecular subtypes based on DNA methylation profiling assays: (i) An integrative analysis of 491 tumors from untreated patients subdivided Subgroups 3 and 4 into eight molecularly distinct subtypes (I-VIII) with specific, albeit somehow overlapping, genetic and transcriptional signatures [7]; (ii) A study conducted on 740 tumors showed that the initial four subgroups can be further subclassified into twelve different molecular subtypes [9]; (iii) A third study identified seven subtypes among 428 primary tumors. These subtypes were validated in an independent cohort consisting of 276 tumors [10]. This further highlights the molecular diversity and complexity within MB.

The 2021 WHO classification of CNS tumors acknowledged the existence of four MB SHH subtypes and eight non-WNT/non-SHH subtypes [11]. These subtypes exhibit distinct clinicopathological characteristics and have diagnostic, prognostic, and predictive impacts on the treatment response (Figure 1). For instance, the SHH-I (or SHH-β) and SHH-II (or SHH-γ) subtypes are predominantly found in children under the age of two [12]. The recognition and characterization of these subtypes should contribute to a deeper understanding of MB and facilitate tailored therapeutic approaches based on specific molecular profiles.

## 2. Metastasis and Recurrence throughout Medulloblastoma Subgroups

The current standard multimodal treatment of MB (surgery, radiotherapy, and chemotherapy) results in a five-year overall survival rate of approximately 70%. However, the chances of cure vary depending on the genetic subgroup [13], the stage of the disease, and the patient metastatic status at the time of diagnosis. Unfortunately, one third of patients do not respond to treatment and experience relapse within two years. Unfortunately, these relapses are often fatal, with patients succumbing within five years of diagnosis [2]. The median survival of relapsed patients is less than one year [14].

The cure rate ranges from 70 to 80% if the tumor remains localized in the cerebellum, compared to 30 to 40% if the disease is metastatic [15,16]. These figures represent overall statistics and do not account for variations observed among different molecular subgroups. Table 1 compiles the molecular and clinical characteristics of the different subgroups, including the proportion of metastasis within each group. 

WNT tumors generally exhibit low rates of metastasis and have a favorable long-term prognosis [17,18,19]. However, WNTβ tumors are more prone to metastasis compared to WNTα tumors, thus highlighting differences in the activation of the WNT pathway between the two subtypes [9].

**Table 1 biology-12-01028-t001:** Molecular and clinical features of medulloblastoma subgroups. Data from [9,12,20,21,22,23].

Subgroup	Subtype	Frequency	Demography	Main Genetic Events	Metastasis Rate	5-Year Overall Survival
WNT	WNTα	70%	Infants-adolescents	*CTNNB1*, *TP53*, *DDX3X*, *MLL2/3* mutation Monosomy chromosome 6	8.6%	97%
	WNTβ	30%	Children–young adults		21.4%	100%
SHH	SHHα	29%	Children-adolescents	Loss of 9q, 10q, 17p *MYCN*, *GLI2*, *YAP1* amp; TP53 mutation	20%	69.8%
	SHHβ	16%	Infants	*PTEN* loss	33%	67.3%
	SHHγ	31%	Infants	Low copy number alterations	8.9%	88%
	SHHδ	24%	Young adults	*TERT* promoter mutation	9.4%	88.5%
Group 3	Group 3α	47%	Infants-children	i17q; loss of 8q and 17p	43.4%	66.2%
	Group 3β	26%	Children-adolescents	*OTX2* gain and *DDX31* loss; activation of *GFI1* and *GFI1B* oncogenes	20%	55.8%
	Group 3γ	28%	Infants-children	i17q; 8q gain and *MYC* amplification	39.4%	41.9%
Group 4	Group 4α	30%	Children-adolescents	i17q; loss of 8p; 7q gain; *MYCN* and *CDK6* amplification	40%	66.8%
	Group 4β	33%	Children-adolescents	i17q; 17p loss; *SNCAIP* duplication	40.7%	75.4%
	Group 4δ	37%	Children-adolescents	i17q; loss of 8p; 7q gain; *CDK6* amplification	38.7%	82.5%

The SHH subgroup can be categorized into four distinct subtypes: α, β, γ, and δ, with different age distributions (Table 1). SHHα is primarily observed in children and is characterized by the following features: *TP53* mutations; focal amplifications in *MYCN*, *GLI2*, and *YAP1*; loss in 9q, 10q, 17p. SHHβ is mainly seen in infants and is associated with a high metastatic rate and PTEN deletion. It has the poorest prognosis [9,12]. SHHγ demonstrates a more favorable outcome while SHHδ is predominantly present in adults and also displays a favorable outcome.

Group 3 is associated with the worst prognosis among all subgroups, and significant differences exist between its subtypes (Table 1). The characteristics of Group 3 subtypes with respect to Group 4 were recently specified further [23]. Sharma et al. suggested that these two groups can be further divided into eight subtypes each with distinct metastatic status and survival outcomes.

Similarly, Group 4 exhibits significant differences in overall survival rates, while the rate of metastasis remains relatively constant across its subtypes (Table 1).

These observations indicate that the rate of metastasis does not necessarily correlate with overall survival in medulloblastoma. This raises the question of whether progression-free survival might serve as a better indicator of MB severity. It also emphasizes the importance of considering the various treatments administered to patients, in addition to the biological and molecular differences among MB. Overall, the complex nature of medulloblastoma underscores the need for a comprehensive understanding of its subtypes, their specific characteristics, and their responses to treatment. This knowledge is crucial for developing more targeted and effective therapeutic approaches for improved patient outcomes. 

## 3. Routes of Metastatic Dissemination in Medulloblastoma

Metastasis is the major cause of mortality in children with MB, mainly because the primary tumor is surgically removed early in the treatment process and because metastases appear very early during the development of the pathology [24]. Among the molecular subgroups of MB, Group 3 tumors are associated with the poorest prognosis and exhibit the highest incidence of metastasis, both at initial diagnosis and recurrence. In contrast, WNT MBs, which represent the subgroup with the most favorable prognosis, demonstrate the lowest incidence of metastasis [25].

Leptomeningeal metastasis refers to the presence of tumor cells in the leptomeninges, which include the arachnoid mater, pia mater, subarachnoid space, and other compartments of cerebrospinal fluid (CSF). In the context of MB, leptomeningeal disease is the predominant pattern of metastasis, and is responsible for nearly 100% of deaths associated with the disease. Tumor cells have a propensity to spread primarily to the spinal and intracranial leptomeninges. This observation strongly supports the hypothesis that MB spreads through the CSF rather than the bloodstream [25,26]. The dissemination of tumor cells within the CSF compartment allows them to access various regions of the central nervous system. The preference for leptomeningeal spread in MB highlights the importance of understanding the mechanisms underlying this specific mode of metastasis. It also underscores the need for effective treatment strategies targeting the elimination or control of tumor cells within the CSF and leptomeninges. However, Garzia and colleagues demonstrated the presence of circulating MB cells (the so-called circulating tumor cells; CTC) in the peripheral blood, thus suggesting that MB cells can undergo hematogenous dissemination [27], leading them towards the leptomeninges where they contribute to the formation of leptomeningeal metastases. The tumor microenvironment, consisting of extracellular signals and cellular components, plays a critical role in facilitating the spread of MB cells. In the hematogenous pathway, the metastatic dissemination of MB is governed by the CCL2/CCR2 signaling axis [27,28,29]. This signaling axis is responsible for orchestrating the process by which MB cells metastasize. Notably, the expression of CCL2 in non-metastatic xenografted cell lines significantly increases metastasis, suggesting a pivotal role for this axis in MB metastatic dissemination [27]. 

These findings underscore the importance of understanding the interplay between tumor cells and their microenvironment in driving the hematogenous spread of MB. Further research in this area is crucial to unravel the intricate molecular mechanisms involved. By elucidating these mechanisms, potential therapeutic targets can be identified to disrupt the metastatic process and improve treatment outcomes for patients with MB. 

It is possible that several different dissemination pathways occur simultaneously for metastatic carcinomas. Indeed, for breast cancer metastasis, both lymphatic and hematogenous dissemination are known to occur [30,31]. Extraneural spread of MB through the lymphatics was suspected as soon as 2009, although, at that time, the presence of lymphatics in the brain was highly controversial [32]. In the CNS, a lymphatic network has been described, particularly in the meninges (within the dura mater), which facilitates CSF drainage (Figure 2). Part of the CSF (in the subarachnoid space) drains into the cervical lymph nodes connecting with the lymphatic circulation [33,34,35]. This finding suggests that leptomeningeal metastasis occurs not only via the CSF, but also through CNS lymphatics. In MB, we recently provided evidence supporting that the VEGFC/VEGFC receptor axes and associated lymphangiogenesis play a subgroup-specific role in the development and aggressiveness of MB [36].

In approximately 7% of cases, MB metastases can spread to the lungs, bones, liver, or lymph nodes [37,38,39]. The spread and development of these metastases are probably facilitated by blood and lymphatic networks.

In rare instances, MB metastases can also be found within the spinal cord (vertebral intramedullary metastases) [40,41]. Since the CSF extends along the spinal cord, it is possible that it serves as the primary transport route for these metastases. Thus, metastatic dissemination may occur through the lymphatic route, responsible for local CNS metastases, as well as the blood route, which may be more associated with distant metastases outside the CNS.

## 4. Tumor Angiogenesis: Scientific Context and Therapeutic Failure

Angiogenesis involves the formation of new blood vessels from pre-existing ones and its balance is maintained by the interplay between pro-angiogenic and anti-angiogenic factors (Figure 2). 

In addition to its role in embryogenesis, angiogenesis contributes to organ growth. In physiological conditions, angiogenesis occurs in a controlled manner during specific events such as tissue repair, gestation, the ovarian cycle, or in response to ischemia (lack of blood supply to tissues) [42]. Pro-angiogenic factors, such as vascular endothelial growth factor (VEGF), promote the growth of new blood vessels, while anti-angiogenic factors, such as thrombospondin-1, inhibit angiogenesis, maintaining a delicate balance. However, during tumor progression and metastatic dissemination, this balance is disrupted, leading to abnormal and dysregulated angiogenesis. Tumors require a blood supply to support their growth and metastasis, and therefore, they stimulate angiogenesis to create new blood vessels that can deliver oxygen, nutrients, and growth factors to the tumor cells. This process allows the tumor cells to establish themselves in new locations and promotes the development of metastases. The dysregulated angiogenesis in cancer is driven by the overexpression of pro-angiogenic factors and the downregulation of anti-angiogenic factors. This imbalance promotes the formation of an abnormal tumor vasculature characterized by leaky, disorganized, and tortuous blood vessels. The abnormal tumor vasculature not only supports tumor growth but also contributes to tumor progression by facilitating intravasation of cancer cells into the bloodstream, leading to distant metastasis.

Understanding the molecular mechanisms underlying angiogenesis and its dysregulation in cancer has led to the development of anti-angiogenic therapies, which aim to disrupt the tumor vasculature and inhibit tumor growth. These therapies target pro-angiogenic factors or their receptors to inhibit the formation of new blood vessels, thereby starving the tumor of its blood supply.

Similar to other solid tumors, MB demonstrated high endothelial cell proliferation and angiogenic activity [43,44,45], thus suggesting that therapeutic strategies targeting the vascular system might be developed efficiently against this pathology. Consequently, the general model described here in the following chapters, might be applicable to medulloblastoma.

### 4.1. Tumor Neovascularization

As a tumor grows, its demand for oxygen and nutrients exceeds what can be supplied by simple diffusion from nearby blood vessels. The tumor then secretes pro-angiogenic signals including VEGFA to stimulate the formation of new blood vessels from existing vessels. Tumor angiogenesis involves the sprouting and remodeling of nearby blood vessels to supply the growing tumor with a network of blood vessels. Tumor blood vessels are often disorganized and abnormal. They are leaky and inefficient, thus leading to inadequate blood flow, poor oxygenation, and uneven distribution of nutrients within the tumor. Once the tumor has established a network of blood vessels, it gains the ability to invade surrounding tissues and spread to distant sites through the bloodstream. By comparing weakly vascularized quiescent tumors and strongly vascularized fast-growing tumors, Folkman established that initiation of tumor angiogenesis is necessary for tumor progression [46]. He also isolated a tumor-produced factor, responsible for tumor associated angiogenesis, which he named TAF (tumor-angiogenesis factor). He suggested that blocking this factor (and thus angiogenesis) could stop tumor growth [47]. These observations paved the way to understanding the activation of tumor angiogenesis, also called the “angiogenic switch”.

In contrast to the physiological vasculature, tumor blood vessels and their endothelial lining have an abnormal architecture. These tumor vessels are disorganized; they do not present the classical artery-capillary-vein hierarchy. They are more dilated and form arteriovenous shunts that leads to unstable blood flow [48,49,50]. They have many branches, irregular diameter and increased permeability to macromolecules leading to higher interstitial pressure and thus edema, fibrosis, inflammation and local microhemorrhages [51]. The endothelial cells lining tumor vessels arise from the proliferation of normal endothelial cells from surrounding the tissue and are structurally abnormal. They have many fenestrations and enlarged cell junctions. They overlap and migrate into the lumen of the vessel. The phospholipids of the inner membrane layer of tumor endothelial cells are disorganized and shifted to the outer membrane. This redistribution of phospholipids is caused by the oxidative stress of the tumor microenvironment and hypoxia [52]. Moreover, these tumor endothelial cells have a high proliferation rate compared to normal endothelial cells [53]. The basal membrane is discontinuous or absent. The tumor endothelium is sparsely covered with morphologically abnormal pericytes, indicating less maturity [54]. Smooth muscle cells, positive for the α-SMA (smooth muscle actin) marker, are reduced in xenograft models of lung carcinoma [55].

The endothelial junctions of tumor vessels are also aberrant and less cohesive: the glioblastoma secretome provides pro-angiogenic and inflammatory signals (here CXCL8), disrupts the junctions formed by VE-cadherin and promotes the permeability of brain endothelial cells [56]. Thus, abnormalities in the structure and composition of tumor vessels combined with a microenvironment rich in pro-angiogenic and inflammatory factors, are responsible for the abnormally high vascular permeability of tumors [57]. These vascular abnormalities create a hostile environment characterized by hypoxia, low pH, inflammation, and high interstitial pressure that select the most aggressive cancer cells. The resulting vascular leakage contributes to the increase in tumor interstitial pressure and causes vascular edema, which limits the delivery of chemo-therapeutic agents and the anti-tumor immune response [51]. Finally, destruction the endothelium promotes intravasation of tumor cells and metastatic dissemination via the bloodstream [58] (Figure 3).

### 4.2. Mechanisms of Angiogenic Hijacking

Angiogenesis rapidly becomes essential for tumor development, which is why the tumor hijacks some physiological mechanisms to its advantage. This angiogenic switch is triggered by several factors. Tumor cells have activating mutations of oncogenes such as the RAS family or inhibitory mutations of tumor suppressor genes such as TP53. These mutations are partly responsible for the dysregulation of physiological angiogenesis: they increase the expression of VEGFA (also inducible by hypoxia (HIF-1α)) and reduce the expression of TSP-1 [48]. Hypoxia induces the overexpression of pro-angiogenic factors, especially VEGFA and PDGF. In addition, defective tumor new vessels create a particular metabolic and immunological microenvironment. Hypoxia, resulting from structural vessel abnormality, reduces the energy metabolism provided by the Krebs cycle, thus leading to an accumulation of succinate. The latter binds to the GPR91 receptor and stimulates the vessel growth [59]. Hypoxia allows the recruitment of bone marrow-derived immune cells to tumor sites, including TAMs, neutrophils, mast cells, and myeloid-derived suppressor cells. These cells release pro-angiogenic signals such as VEGFA or MMPs [60] and participate in immune tolerance (Figure 4). Thus, in addition to the secretion of pro-angiogenic factors by the cancer cells, the microenvironment increases their aggressiveness and angiogenesis.

The endothelial lining of tumor blood vessels arises from the proliferation of normal endothelial cells from the surrounding tissue. Thus, the change in phenotype of these endothelial cells, in a tumor context, is due to the microenvironment and to epigenetic factors. In mouse models, identical human colon adenocarcinoma tumors implanted in the liver or skin show different tumor endothelial phenotypes. The vessels of the hepatic tumor are narrower and more permeable than those of the subcutaneous tumor. The number of leukocytes in the liver tumor and the amount of VEGFA mRNA are decreased compared to the subcutaneous tumor [61]. The phenotype of tumor vessels generated during cerebral or subcutaneous implantation of human glioblastoma, rhabdomyosarcoma and murine mammary carcinoma tumors differs depending on the tissue [62]. These two studies demonstrate the role played by the microenvironment of the receiving tumor tissue. Tumor cell conditioned medium leads to epigenetic alteration of gene expression in cultured endothelial cells [63,64]. The gene expression profiles in primary cultures of isolated glioblastoma endothelial cells differ from those of endothelial cells from normal brain tissue [65,66]. The epigenetic profile of tumor endothelial cells is thus influenced by the tumor environment. In addition, the endothelial cells of tumor blood vessels are genetically reprogrammed. Endothelial cells isolated from human melanoma and liposarcoma xenografts exhibit aneuploidy and multiple centrosomes. These CD31+ cells express lower levels of TIE1 and TIE2, proliferate faster, have lower serum requirements, and are more sensitive to FGF and EGF than normal endothelial cells. These important findings suggest that genetic alterations of tumor endothelial cells influence the cell phenotype [67]. Tumors can take up endothelial cells from existing blood vessels and change the phenotype of the endothelium [68]. In some tumors, the vessels are lined by tumor cells instead of endothelial cells. In glioblastoma, a significant proportion of the endothelial cells associated with the tumor vessels are of neoplastic origin. Neural stem cells of glioblastoma promote angiogenesis by releasing VEGFA and differentiating into a tumor endothelial phenotype. These cells connect to tumor vessels and the resulting hybrid vessels are functional [69]. This differentiation mechanism, termed vascular mimicry, is unclear, but the presence of intravascular tumor cells interferes with targeted anti-angiogenic therapies [48,70].

### 4.3. Anti-Angiogenic Therapies and Their Limits

Over the past decades, angiogenesis has emerged as a critical strategy in oncology. The aim of anti-angiogenic therapy is to starve tumors by disrupting their oxygen and nutrient supply, ultimately reducing tumor proliferation. Angiogenesis is tightly regulated by a delicate balance of activating and inhibiting signals, as discussed earlier. However, in tumor tissue, VEGFA_165_ is often overexpressed as the main vascular growth factor. VEGFA_165_ promotes angiogenesis and tumor growth by binding to and activating VEGFR1 and VEGFR2 thereby initiating a cascade of signaling events. To counteract these effects, a range of anti-VEGFA_165_/VEGFRs agents have been developed. They demonstrate potent efficacy in inhibiting angiogenesis and suppressing tumor growth in preclinical models. As a result, several anti-VEGFA_165_/VEGFRs have gained approval for the treatment of various cancers. In MB, VEGFA was demonstrated as a potent biomarker and prognosis marker, especially in the SHH subgroup [71].

Bevacizumab (BVZ, Avastin^®^) is a humanized monoclonal antibody directed against biologically relevant VEGFA isoforms. It is commonly used as a standalone therapy or in combination with chemotherapy in colorectal and ovarian cancer. The addition of BVZ to chemotherapy improves patient survival and response rates in colon [72] and ovarian [73] cancers. In some cases of glioblastomas, BVZ is used as monotherapy. It was also combined with IFN-α-2a for the treatment of metastatic renal carcinomas, [74,75], but its use in this context is no longer prevalent. BVZ is now used for renal cancer in combination with atezolizumab (Tecentriq^®^), an anti-PDL1 antibody [76]. BVZ was used with success on MB [77]. Other compounds target the tyrosine kinase activity of VEGFA receptors. Sunitinib (Sutent^®^) inhibits VEGFR1, VEGFR2 and VEGFR3, as well as other receptors with tyrosine kinase activity (PDGFR, CSFR1, c-KIT, etc.), downstream of the signaling of several pro-angiogenic factors. It is approved for the treatment of metastatic renal cell carcinoma and advanced neuroendocrine pancreatic cancer. Other multi-kinase treatments such as pazopanib (Votrient^®^), vandetanib (Caprelsa^®^) or sorafenib (Nexavar^®^) are used for the treatment of various metastatic cancers (kidney, thyroid, liver, etc.).

Despite an initial period of clinical benefit with improved progression-free survival and tumor regression, none of these treatments resulted in complete cure. The treated primary tumors relapse and persistent malignant cells proliferate and disseminate in distant healthy tissues, giving rise to metastases. The mechanisms of resistance in tumors are not fully understood to date. They involve events related to the tumor microenvironment, intrinsic resistance associated with the redundancy of pro-angiogenic factors and acquired resistance leading to tumor revascularization.

Tumor cells use alternative pro-angiogenic factors independent of the VEGFA/VEGFR pathway to resist conventional anti-angiogenic therapies. When the VEGFA/VEGFR pathway is blocked, other pro-angiogenic factors restore tumor angiogenesis. In experimental models of pancreatic cancer in mice, antibodies blocking the VEGFR2 receptor initially inhibit tumor growth. At an advanced stage of the disease, the tumors become resistant and progress. This resistance is correlated with the overexpression of FGF1 and FGF2. However, tumors treated with an FGF inhibitor alongside the VEGFR2 inhibitor show reduced revascularization and tumor progression compared to tumors treated with the VEGFR2 inhibitor alone [78]. These findings highlight the importance of targeting multiple pro-angiogenic pathways to overcome resistance and improve treatment outcomes.

Kidney cancer cells overexpress several redundant pro-angiogenic factors, including VEGFA and the cytokine CXCL8, compared to healthy tissues. In cellular models of kidney cancer, BVZ traps VEGFA and increases the compensatory production of pro-angiogenic ELR+CXCL cytokines. In experimental kidney cancer in mice, anti-VEGFA accelerates tumor growth but when combined with an anti-CXCL8 antibody, tumor growth is inhibited. In animals treated with BVZ, the density of tumor blood vessels is decreased while the density of tumor lymphatic vessels is increased. This phenomenon is accompanied by an increase in the levels of the main lymphangiogenic factors VEGFC and VEGFD. Conversely, treatment with anti-CXCL8, either alone or in combination with BVZ, reduces the levels of VEGFC. Furthermore, the expression of the receptor-type protein-tyrosine phosphatase-κ (RPTP-κ), an inhibitory EGF receptor (EGFR) phosphatase, is decreased in cells from BVZ-treated tumors [79].

MBs, like kidney cancer, are highly vascularized tumors that overexpress several members of the VEGF family and many other markers of angiogenesis (VEGFB, VEGFC, FGF, angiopoietin). However, the response rate of anti-angiogenic treatments in these tumors is low, mainly because of the redundancy of angiogenic factors. Furthermore, these treatments can have detrimental effects on the development of children making their use challenging [80]. Recent experiments have demonstrated the potential of BVZ in the treatment of pediatric MB patients who experience relapse. When used in combination with metronomic chemotherapy, BVZ has shown promise in improving outcomes for these patients [81]. However, despite these positive results, BVZ in this context is not a curative chemotherapy. Another multi-kinase inhibitor, axitinib (Inlyta^®^), has shown relevant effects on the development of experimental MBs in mice [82,83].

The evasion of tumor from therapies targeting the VEGFA/VEGFR axis may be attributed to several factors: increased activation of EGFR; the development of lymphangiogenesis which provides an additional route for metastatic dissemination; the production of compensatory pro-angiogenic factors (FGF, VEGFB, angiopoietins, cytokines of the ELR+CXCL family) to counteract the effects of VEGFA inhibition (Figure 5). 

Most anti-angiogenic approaches focus on inhibiting the signaling pathway of pro-angiogenic factors, either by interacting with the factor itself (e.g., BVZ, the anti-VEGFA) or by blocking its receptor on the tumor cell surface (e.g., sunitinib, the anti-VEGFR). However, tumor cells have an arsenal of strategies to evade the effects of anti-angiogenics. Particularly, it was shown that glioblastoma patients inevitably develop resistance mechanisms to BVZ, by activating alternative pathways leading to neoangiogenesis [84]. In consequence, due to the limited efficacy of these anti-angiogenic therapies in certain types of cancers including MB, there is a need for the development of new strategies to overcome resistance and improve treatment outcomes.

### 4.4. Novel Anti-Angiogenic Therapeutic Approaches

An alternative approach to conventional chemotherapy is metronomic chemotherapy [85,86,87], which aims at administrating chemotherapeutic agents at relatively low, minimally toxic doses and with no prolonged drug-free breaks [88]. Interestingly, Browder et al., demonstrated that cyclophosphamide-resistant leukemia can be killed in vivo by metronomic doses of this drug, and that this dosing schedule inhibits tumor growth primarily through antiangiogenic mechanisms [89]. A combinatorial metronomic antiangiogenic clinical study (MEMMAT; ClinicalTrials.gov Identifier: NCT01356290) demonstrated an increase in the median OS of the patients with MB [81], thus confirming the relevance of this protocol to treat MB. 

Since MB is such a highly vascularized tumor, targeting blood vessels and angiogenesis seems a promising therapeutic strategy: the local addition of angiogenesis inhibitors should lead to significant prolongation of patient survival. Several types of strategies have been attempted in this regard.

Oncolytic virotherapy represents a promising therapeutic strategy. Oncolytic viruses, especially measles viruses, have been developed to selectively infect and kill tumor cells, thus inhibiting tumor growth, while leaving the normal surrounding tissue unaffected [90]. Oncolytic measles viruses have been efficiently used against MB, in orthotopic mouse models of localized and disseminated disease [91,92]. Moreover, Hutzen et al. demonstrated that engineered oncolytic measles viruses expressing both endostatin and angiostatin, two inhibitors of angiogenesis, prevented the secretion of several angiogenesis factors in vitro, and inhibited endothelial cell tube formation [93]. Oncolytic measles viruses thus exhibit a potential therapeutic benefit for MB patients, that remains to be demonstrated.

Several pieces of evidence have been collected regarding the aberrant expression of miRNAs in various tumors [94]. miRNAs control tumor development genes, regulate apoptosis and proliferation of tumor cells, monitor the response to DNA damage and repair, and response to hypoxia, as well as interaction of the tumor with its microenvironment [94,95]. Many miRNAs are involved in signaling pathways associated with cell cycle regulation and apoptosis. In particular, miR-221-3p reduces MB cell proliferation by inducing apoptosis and G0/G1 arrest by suppressing eukaryotic translation initiation factor 5A-2 (EIF5A2) [96]. Moreover, miRNAs also contribute to the regulation of the tumor vascular microenvironment by controlling angiogenesis. Especially, in MB, a critical role for miR-494 was found in the suppression of tumor angiogenesis, which is translated into increased radiosensitivity of MB cells [97]. MiRNAs might thus be part of the future therapeutic arsenal against MB.

The presence of physiological barriers, primarily the blood-brain barrier (BBB), is the main limitation to the efficient drug delivery to the brain. BBB hampers the efficacy of several drug therapies. The development of nanomedicine, i.e., the use of nanotechnology with medical purposes, appears as a strong ally. Nanomedicine can increase the intracellular levels of a drug by encapsulating it in different nanocarriers that will bypass the BBB, the cell efflux pumps, when administrated systemically. They can even be applied locally at the time of surgery. If several new nanoparticles have already been experimented in the case of glioblastoma, medulloblastoma treatment is still elusive. Further characterization of the BBB is expected to progress in that matter and help personalize the nanoparticle form of chemotherapeutic treatment against MB.

## 5. From Molecular Pathology to Targeted Therapies

The current approach to cancer treatment lacks the ability to adequately address the inter-tumor heterogeneity observed in different subgroups and subtypes of patients. Therapeutic approaches must therefore be further developed in order to fight tumors more specifically without affecting healthy tissue. The previously described molecular definition of subgroups and subtypes paved the way for the development of more specific therapies in the era of personalized medicine (Figure 1; Table 1 and Table 2).

### 5.1. WNT Subgroup Medulloblastomas

WNT MBs have a looser BBB than other MBs. This feature contributes to the ability of chemotherapy to penetrate the CNS and these patients have a good prognosis [16]. The WNT pathway plays an important role in tissue regeneration and bone repair during development [98]. Therefore, therapies targeting the WNT pathway are not currently developed for MB. Due to the good prognosis of these patients, clinical trials (NCT02724579, NCT02066220 and NCT01878617) are ongoing that aim to reduce the dose of radio- and chemotherapy (therapeutic de-escalation) and improve the quality of life of these patients without compromising their prognosis.

### 5.2. SHH Subgroup Medulloblastomas

Constitutive activation of the SHH pathway is ripe to the development of inhibitors of this pathway. In particular, smoothened (SMO), the main activating ligand, has been targeted. However, inhibitors such as vismodegib (Erivedge^®^) or sonidegib (Odomzo^®^) are ineffective if the tumors carry mutations of effectors of the SHH pathway (such as SUFU or GLI mutations [99,100]). In addition, vismodegib causes bone and dental problems, disproportionate growth and precocious puberty that persist long after stopping treatment [101]. Stratification of patients at diagnosis is therefore an essential prerequisite to avoid unnecessary side effects in patients who do not respond due to the genetic characteristics of their tumor. Pre-clinical efforts to compensate for these resistances are ongoing. Itraconazole and arsenic trioxide, two agents in clinical use, inhibit SMO activity and certain GLI mutations implicated in vismodegib and sonidegib resistance. Itraconazole targets the intracellular part of SMO, and arsenic trioxide inhibits GLI2. These inhibitors, alone or in combination, inhibit the growth of SHH MB and prolong the survival of naive or SMO inhibitor-resistant mice [102]. These inhibitors are FDA-approved, and their toxicity is characterized. Their repositioning in the MB SHH would thus be facilitated and represents hope for patients with mutations downstream of SMO.

### 5.3. Group 3/4 Medulloblastomas

The limited understanding of tumorigenesis in these subgroups limits the development of new targeted therapies. However, there is an urgent need to offer patients new alternatives. One of the first ways is to tailor treatments to patients’ risk. The NCT01878617 clinical trial proposes pemetrexed and gemcitabine for newly diagnosed intermediate- and high-risk patients who have first received radiotherapy and standard chemotherapy. For relapsed MB, prexasertib (inhibitor of checkpoint kinase-1 and -2 proteins involved in cell cycle regulation) is offered in combination with cyclophosphamide (trial NCT04023669) [103].

However, these strategies are not specific to these subgroups. Targeting MYC, which is frequently mutated in these patients, would be particularly important. The bromodomain inhibitor JQ1 blocks MYC activity in mice [104]. Currently, no clinical trials are being conducted in MB. Other bromodomain inhibitors (CPI-0610 and MK-8628) are in phase 1 clinical trials in hematological malignancies, prostate cancer, breast cancer, non-small cell lung cancer and glioblastoma.

MBs with MYC mutations appear to rely on CDK4/6 for their proliferation. In Group 3, palbociclib, a potent CDK4/6 inhibitor in a mouse model [105], has been clinically tested in MB and other brain tumors (NCT02255461). This drug is FDA-approved for breast cancer. However, this treatment does not seem to benefit patients with MB [106].

The development of new, more targeted alternatives for these subgroups of patients remains urgent, given their unfavorable prognosis.

It is important to notice that, given the significant efficacy of current therapies and the lack of prospects for these new targeted therapies, the latter are only available when the patients relapse, regardless of the MB subgroup. Moreover, they may have fewer side effects for young patients. 

Unfortunately, the number of targeted therapies currently on the market is yet insufficient, despite considerable preclinical efforts. These therapeutic approaches need to be further developed to achieve better efficacy of treatments by reducing their toxicity.

## 6. What about Immunotherapies?

Cancer immunotherapy aims to stimulate and improve the patient’s anti-tumor immune response by preserving healthy tissue. It is now considered a new pillar of cancer treatment and is already used clinically for the treatment of lung, melanoma, and kidney cancers. However, its use in brain tumors is not yet well known. There are different types of immunotherapies, e.g., immune checkpoint inhibitors, natural killer (NK) cells, CAR-T cells (Chimeric Antigen Receptor T cell), cancer vaccines, oncolytic viruses and immunomodulators (e.g., cytokines or antibodies) [107] (Figure 1; Table 2).

### 6.1. WNT Subgroup Medulloblastomas

To escape the response of T lymphocytes, cancer cells up-regulate the expression of immune checkpoints (e.g., PDL1 or B7). This immunosuppressive mechanism, hijacked by the tumor cells, is designed to attenuate the immune response, and prevent autoimmunity. Inhibition of these checkpoints enables reactivation of the antitumor immune system.

Among the best-known immune checkpoints, the binding of PDL1 (Programmed cell death ligand 1) to its receptor PD1 (Programmed cell death protein 1) plays an important role. PDL1 is expressed by tumor cells, binds to its PD1 receptor on T lymphocytes (LT), by which it prevents activation of the LT. PD1 is also expressed by B cells (B cells), activated monocytes, dendritic cells, and NK cells [108]. This binding therefore results in extinction of the antitumor reaction. The PD-1/PD-L1 pathway is at the heart of tumor escape from the immune system and, as such, represents a potential therapeutic target.

Overall, MBs express low levels of PDL1. However, this expression correlates with reduced infiltration of CD8+ T cells and poor prognosis [109]. PDL1 expression seems to be related to the MB subgroups. SHH tumors express high levels, whereas Groups 3 and 4 display low levels. An inflammatory microenvironment is required to induce PDL1 expression in these tumors. Interferon gamma (IFN-**γ**) stimulates TH1 cytokines, which induce PDL1 expression in MBs of the SHH subgroup [110]. The efficacy of anti-PDL1 in mice depends on the timing of treatment. They are effective from day 7 after transplantation but are ineffective at the time of tumor inoculation. In mice, SHH tumors have a distinctly inflammatory microenvironment compared to the tumors of Groups 3 and 4. Nevertheless, Group 3 tumors respond better to anti-PD1 due to greater infiltration of PD1+ CD8+ T cells and increased survival [111]. The lack of response to anti-PD1 in SHH tumors may be related to the presence of MDSC (Myeloid-Derived Suppressor Cells) and TAM (Tumor Associated Macrophages) involved in immune tolerance via inhibition of the T response [111,112]. Group 3 patients would therefore respond better to immunotherapy, while SHH patients could be resistant to anti-PD1 therapies. The genetic subgroup, the tumor microenvironment and the timing of administration are thus parameters that need to be considered to improve the effect of this therapy.

Clinical trials are currently underway in MB and other CNS tumors. The NCT02359565 phase I clinical trial is investigating the efficacy of the anti-PD1 monoclonal antibody, pembrolizumab (Keytruda^®^) in children and young adults with recurrent brain tumors including MB. A phase II clinical trial (NCT03173950) is investigating the efficacy of nivolumab (Opdivo^®^), another anti-PD1 monoclonal antibody, in adults. Finally, the NCT03130959 phase II clinical trial is investigating the effect of nivolumab, alone or in combination with ipilimumab (or Yervoy^®^, an anti-CTLA4).

The CTLA4 receptor (Cytotoxic T-Lymphocyte Antigen 4 protein) is also an immune checkpoint. It is expressed by LTs and its interaction with B7 (on antigen-presenting cells or tumor cells) transduces an inhibitory signal to lymphocytes. Inhibition of this binding is also a therapeutic way to reactivate the antitumor immune system. B7-H3, a glycoprotein of the B7 family, is overexpressed in all subgroups of MB [113]. Two ongoing clinical trials (NCT04167618 and NCT04743661) are investigating the effect of radiotherapy coupled with anti-B7-H3 immunotherapy (radiolabeled monoclonal antibody) intrathecally in MB [114].

The use of immune checkpoint inhibitors is poorly documented in MB. However, it may be held out as hope, given the urgency to develop less toxic alternative therapies for patients. 

Preclinical studies show that the efficacy of new potential treatments must be evaluated and optimally adapted depending on the genetic subgroup of MB. However, in the current clinical trials, this stratification is not performed. In addition, the small number of MB models in immunocompetent mice limits these types of studies.

### 6.2. Natural Killer NK Cells

NK cells are cytotoxic lymphocytes that are able to recognize and lyse damaged “self” and “non-self” cells such as tumor cells. This lysis occurs via the perforin/granzyme and the IFN**γ** pathways. NK cells express germline-encoded activating and inhibitory receptors. Inhibitory receptors (KIR, Killer cell Ig-like Receptors) recognize major histocompatibility complex class I (MHC-I) and make NK cells resistant to healthy tissue and self-proteins. Activation receptors recognize activating signals generated by damaged, infected, or cancerous cells. They also secrete inflammatory and immunosuppressive cytokines that control the adaptive immune response [115].

Autologous NK cells can be harvested, activated, propagated, and genetically modified to increase their antitumor activity, and then returned to the patient. This technique is particularly effective in hematological malignancies and solid tumors [116].

MB cell lines express NK-activating ligands. In particular, the Daoy cell line has a high level of NKG2D ligands, an activating NK receptor, responsible for its lytic activity. This property leads to the lysis of MB cells by activated human NK cells in vitro. The lysis is independent of the presence of the CD133 stemness marker [117]. Moreover, NK cells induce apoptosis of human MB tumor cells in the cerebellum of immunodeficient NSG mice. These tumors show a decreased expression of MHC-I, making them more sensitive to lysis by NK [118]. NK-based immunotherapy is therefore an effective approach both on the primary tumor and on the tumor stem cells responsible for self-renewal. However, MB cells also generate immunosuppressive signals (such as TGF-β), so their elimination by NKs is incomplete. Creating NK cells that express a dominant-negative TGF-β allows this inhibiting signal to be ignored [119]. Genetically modified NKs therefore present an advantage and guide the development of an immune strategy.

A phase 1 trial (NCT02271711) is currently attempting to evaluate the efficacy of autologous NK cells activated ex vivo (with artificial antigen-presenting cells) and injected into the cerebellum of children with relapsed or refractory disease [103]. The outcome of this trial will provide a first insight into the effect of NK immunotherapy in patients.

### 6.3. CAR-T Cells

CAR-T cell therapy consists of:-Collecting autologous or allogeneic T cells by apheresis and genetically modifying them to express tumor antigen-specific receptors (CARs) by viral transduction;-Amplifying these LTs and reinjecting them into the patient after lymphodepletion, which promotes the expansion and persistence of the CAR-Ts. Thus, these CAR-Ts enable a specific immune response against cancer cells.

The use of CAR-Ts has shown efficacy in hematological malignancies (especially with anti-CD19/CD20 CAR-Ts). However, one of the main side effects is cytokine release syndrome associated with reversible neurological damage after treatment [120,121]. Their efficacy is controversial in solid tumors. Extensive preclinical studies are needed to identify tumor-specific antigens that are not expressed by healthy tissue.

In MB, one of the interesting targets is the receptor tyrosine kinase ERBB2 (HER2). Although all MBs present ERBB2 mRNA, the HER2 protein is expressed by 40% of MBs and is associated with a poor prognosis [122]. However, this receptor is undetectable in normal developing cerebellum [123,124]. It is therefore a relevant target for the development of CAR-Ts. Low-dose anti-HER2 CAR-Ts lead to rapid regression of experimental MB in mice. These immunodeficient mice, treated with CAR-Ts directly in the cerebellum show no systemic toxicity [125]. A phase 1 clinical trial is currently ongoing (NCT03500991) to investigate the effect of anti-HER2 autologous anti-HER2 CD4+ and CD8+ CAR-Ts in patients with relapsed/refractory HER2-positive MB.

Similarly, EPHA2, another tyrosine kinase, and IL13Rα2 targets are expressed by MBs (and ependymomas), but not by the developing healthy brain. CAR-T EPHA2, HER2 and IL13Rα2 cells, alone or in combination with the CSF are effective against primary, metastatic, and recurrent Group 3 MB in mouse models (as well as ependymomas) [126].

Adoptive LT therapy is therefore very promising for the treatment of MB. This therapy is tailored to each patient and LTs can pass through the BBB and infiltrate the brain [114]. However, tumors can negatively regulate the antigenic target and patients can develop resistance, as in leukemia [127]. To overcome antigenic escape, multi-variant CAR-Ts can be developed, as previously described for EPHA2, HER2 and IL13Rα2 CAR-Ts. In addition, adverse effects on MB are not known, so implementation of CAR-T strategy should be considered with caution.

## 7. Conclusions

In conclusion, a significant proportion of patients with MB remain incurable, despite ongoing clinical trials. One of the reasons lies in the heterogeneity of the disease, and the current non-targeted treatments (radio and chemotherapy) which lead to the selection of the most aggressive cells responsible for resistance. Relapse is associated with a poor outcome in metastatic patients. No targeted treatment is proposed to these relapsed patients.

As developed in the course of this review paper, the hematogenous route constitutes the main route for the dissemination of distant metastases. Consequently, there are several promising avenues for future exploration. First, further investigation is warranted to elucidate the underlying mechanisms of angiogenesis in medulloblastoma, as this would facilitate the development of more targeted and effective therapies. Second, clinical trials focusing on anti-angiogenic agents, either as monotherapies or in combination with existing treatments, should be pursued to evaluate their efficacy and safety profiles. Moreover, novel medulloblastoma-specific angiogenic targets must be identified in order to pave the way for the development of innovative therapeutic strategies.

Furthermore, it is crucial to establish robust preclinical models that accurately recapitulate the complex tumor microenvironment of medulloblastoma. These models can help in the evaluation of anti-angiogenic therapies and provide valuable insights into their potential clinical translation. Additionally, exploring the role of angiogenesis in medulloblastoma subtypes and investigating potential subtype-specific therapeutic approaches is an avenue worth exploring. In conclusion, while anti-angiogenic therapies hold promise for medulloblastoma treatment, further research is necessary to overcome existing challenges and optimize their clinical application. By addressing these future directions, we can progress in this field and potentially enhance therapeutic outcomes for patients with medulloblastoma.

In parallel, while various therapeutic approaches have been developed, understanding the mechanisms of lymphatic system metastasis in MB could significantly enhance the effectiveness of immune-based therapies such as immune checkpoints and CAR-T cell therapy. Targeting the lymphatic system and its interactions with immune cells may improve immune responses against MB and reduce the incidence of metastases.

Further investigations are necessary to fully understand the complexity of the lymphatic system in MB. By expanding our knowledge in this area, we can potentially enhance treatment strategies and improve outcomes for patients with MB.

In addition to relapses, a major problem for young patients with MB is that standard treatments induce significant side effects that affect their quality of life. The doses of radio and chemotherapy are not adapted according to the subgroup. De-escalation studies of doses should be carried out especially in patients of the WNT group with a good prognosis. In childhood cancers, and in particular in MB, metronomic chemotherapy (at lower doses and continuously) needs to be developed. Although the mechanisms of action remain to be elucidated, these treatment profiles adapted to children with MB would reduce toxicities and side effects.

## Figures and Tables

**Figure 1 biology-12-01028-f001:**
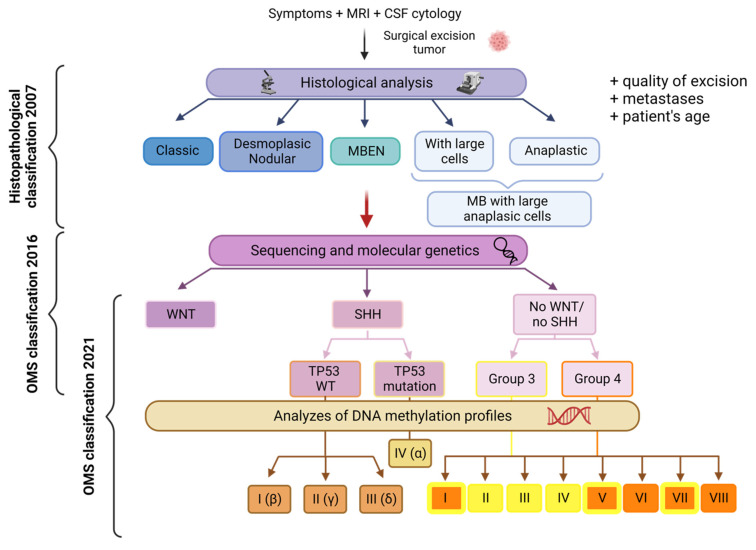
Representation of the main MB classifications associated with the techniques that led to their discovery. This figure provides an overview of the histopathological classification of MB. It encompasses the ancestral classification, the 2016 WHO classification with four subgroups, and the updated 2021 WHO classification with 12 subtypes (four subtypes for the SHH Group and eight subtypes for the non-WNT/non-SHH Group). The subtypes belonging to Group 3 are represented in yellow, while those from Group 4 are depicted in orange. Notably, subtypes I, V, and VII display characteristics of both Group 3 and Group 4. MRI: Magnetic Resonance Imaging; CSF: Cerebrospinal Fluid; MBEN: Medulloblastoma with Extensive Nodularity. Created with BioRender.com, accessed 1 June 2022.

**Figure 2 biology-12-01028-f002:**
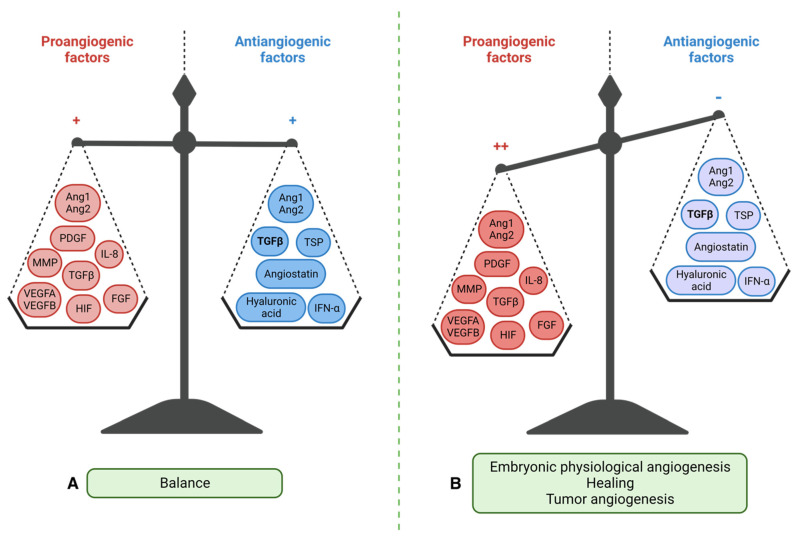
Control of angiogenesis by pro- and anti-angiogenic factors. (**A**) Physiological angiogenesis: balance between pro-angiogenic factors (VEGFA, VEGFB, HIF, FGF, low concentration TGF**β**, MMP, PDGF, IL-8, Ang1 and 2) and anti-angiogenic factors (derived from hyaluronic acid, angiostatin, INF-**γ**, thrombospondin (TSP), high concentration TGF**β** and Ang1 and 2). (**B**) Activation of angiogenesis by overexpression of pro-angiogenic factors (++) and repression of anti-angiogenic factors (−): case of embryogenesis, healing, or cancers. VEGF = Vascular Endothelial Growth Factor, HIF = Hypoxia-Inducible Factor, FGF = Fibroblast Growth Factor, TGF**β** = Transforming Growth Factor Beta, MMP = Metalloproteinases, PDGF = Platelet-Derived Growth Factor, IL-8 = Interleukin-8, Ang1 and 2 = Angiopoietin1 and 2, INF-**γ** = Interferon-gamma. Created with BioRender.com, accessed 1 June 2022.

**Figure 3 biology-12-01028-f003:**
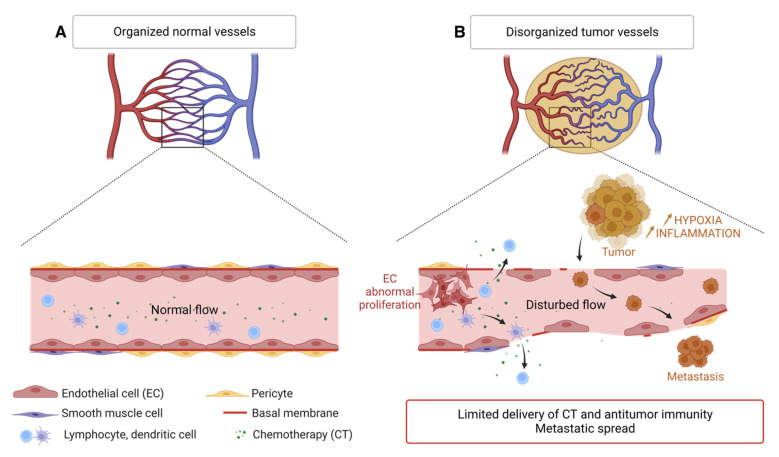
Disorganization of the tumor vascular network compared to the normal vasculature. (**A**) Normal blood vessel with contiguous endothelial cells, a continuous basement membrane, and a layer of smooth muscle cells and pericytes. The circulation of immune cells and macromolecules is normal. (**B**) Tumor blood vessel showing many branches, irregular vessels, interrupted endothelial cells (EC) and basement membrane, few pericytes and smooth muscle cells, causing limited supply of chemotherapy molecules (CT) and antitumor immune cells but favoring the spread of tumor cells. Created with BioRender.com, accessed 1 June 2022.

**Figure 4 biology-12-01028-f004:**
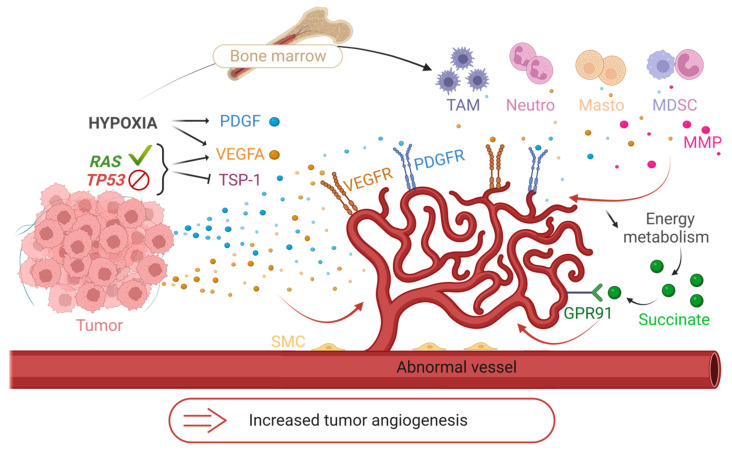
The mechanisms of the angiogenic switch. Illustration of some mechanisms hijacked by tumors to increase their angiogenesis. Release of pro-angiogenic factors (VEGFA, PDGF) and suppression of anti-angiogenic factors (thrombospondin-1; TSP-1) under the influence of genomic instability of tumors (activating RAS mutation or inhibiting TP53) and hypoxia. Recruitment of immune cells from the bone marrow to tumor sites and secretion of pro-angiogenic factors (VEGFA, MMP) under hypoxia. Induction of hypoxia by abnormal tumor vessels and decrease in energy metabolism. Development of angiogenesis by accumulation of succinate and binding to its GPR91 receptor. TAM = Tumor Associated Macrophages, Neutro = Neutrophils, Masto = Mast Cells, MDSC = Myeloid Derived Suppressive Cells, MMP = Metalloproteinases, SMC = Smooth Muscle Cells. Created with BioRender.com, accessed 1 June 2022.

**Figure 5 biology-12-01028-f005:**
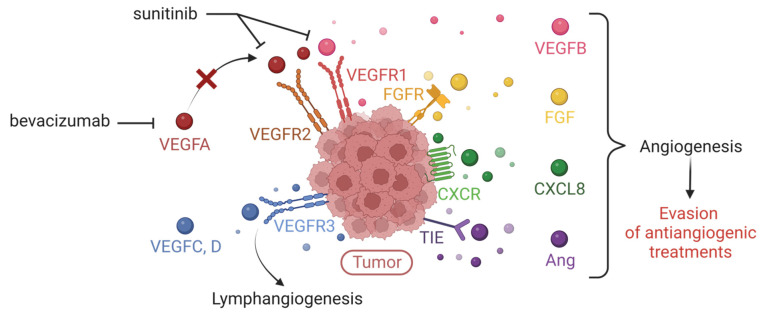
Expression of multiple tumor pro-angiogenic factors responsible for evasion of anti-VEGFA/VEGFR treatments. Illustration of the expression of pro-angiogenic factors that compensate VEGFA by the tumor, thus explaining the limitation of conventional therapies (such as bevacizumab or sunitinib) targeting the VEGFA/VEGFR axis (VEGFR = Vascular Endothelial Growth Factor Receptor). These redundant factors are the other members of the VEGF family, Fibroblast Growth Factor (FGF), the cytokine CXCL8, angiopoietins (Ang). Created with BioRender.com, accessed 1 June 2022.

**Table 2 biology-12-01028-t002:** Clinical trials referencing new treatments under study against pediatric MB.

Reference	Title	Phase	Enrollment	Intervention/Treatment Type
NCT00602667	Risk-Adapted Therapy for Young Children with Embryonal Brain Tumors, Choroid Plexus Carcinoma, High Grade Glioma or Ependymoma	2	293	Drug: Induction Chemotherapy Drug: Low-Risk Therapy Drug: High-Risk Therapy Drug: Intermediate-Risk Therapy
NCT01878617	A Clinical and Molecular Risk-Directed Therapy for Newly Diagnosed Medulloblastoma	2	660	Radiation: Craniospinal Irradiation with boost to the primary tumor site Drug: Cyclophosphamide Drug: Cisplatin Drug: Vincristine Drug: Vismodegib Drug: Pemetrexed Drug: Gemcitabine Other: Aerobic Training Other: Neurocognitive Remediation
NCT02017964	Combination Chemotherapy in Treating Younger Patients With Newly Diagnosed, Non-metastatic Desmoplastic Medulloblastoma	2	26	Drug: Carboplatin Other: Cognitive Assessment Drug: Cyclophosphamide Drug: Etoposide Other: Laboratory Biomarker Analysis Drug: Methotrexate Drug: Vincristine Sulfate
NCT02066220	International Society of Paediatric Oncology (SIOP) PNET 5 Medulloblastoma	2; 3	360	Radiation: Radiotherapy without Carboplatin Drug: Reduced-intensity maintenance chemotherapy Radiation: Radiotherapy with Carboplatin Drug: Maintenance chemotherapy Radiation: WNT-HR < 16 years Radiation: WNT-HR >= 16 years Drug: Induction Chemotherapy Radiation: SHH-TP53 M0 Radiation: SHH-TP53 M+ (germline) Radiation: SHH-TP53 (somatic) Drug: Vinblastin Maintenance
NCT02238899	Multicenter Register for Children and Young Adults With Intracranial Localized Medulloblastoma, CNS-PNET or Ependymoma		354	
NCT02255461	Palbociclib Isethionate in Treating Younger Patients With Recurrent, Progressive, or Refractory Central Nervous System Tumors	1	35	Drug: palbociclib isethionate Other: pharmacological study Other: laboratory biomarker analysis
NCT02271711	Expanded Natural Killer Cell Infusion in Treating Younger Patients With Recurrent/Refractory Brain Tumors	1	12	Other: Laboratory Biomarker Analysis Biological: Natural Killer Cell Therapy
NCT02359565	Pembrolizumab in Treating Younger Patients With Recurrent, Progressive, or Refractory High-Grade Gliomas, Diffuse Intrinsic Pontine Gliomas, Hypermutated Brain Tumors, Ependymoma or Medulloblastoma	1	110	Procedure: Diffusion Tensor Imaging Procedure: Diffusion Weighted Imaging Procedure: Dynamic Contrast-Enhanced Magnetic Resonance ImagingProcedure: Dynamic Susceptibility Contrast-Enhanced Magnetic Resonance Imaging Other: Laboratory Biomarker Analysis Procedure: Magnetic Resonance Spectroscopic Imaging Biological: Pembrolizumab Procedure: Perfusion Magnetic Resonance Imaging
NCT02724579	Reduced Craniospinal Radiation Therapy and Chemotherapy in Treating Younger Patients With Newly Diagnosed WNT-Driven Medulloblastoma	2	45	Drug: Cisplatin Drug: Cyclophosphamide Other: Laboratory Biomarker Analysis Drug: Lomustine Radiation: Radiation Therapy Drug: Vincristine Drug: Vincristine Sulfate
NCT03130959	A Study to Evaluate the Safety and Efficacy of Nivolumab Monotherapy and Nivolumab in Combination With Ipilimumab in Pediatric Participants With High Grade Primary Central Nervous System (CNS) Malignancies (CheckMate 908)	2	166	Biological: Nivolumab Biological: Ipilimumab
NCT03500991	HER2-specific CAR T Cell Locoregional Immunotherapy for HER2-positive Recurrent/Refractory Pediatric CNS Tumors	1	48	Biological: HER2-specific chimeric antigen receptor (CAR) T cell
NCT04023669	Evaluation of LY2606368 Therapy in Combination With Cyclophosphamide or Gemcitabine for Children and Adolescents With Refractory or Recurrent Group 3/Group 4 or SHH Medulloblastoma Brain Tumors	1	21	Drug: Prexasertib Drug: Cyclophosphamide Drug: Gemcitabine Biological: filgrastim Biological: peg-filgrastim
NCT04743661	131I-Omburtamab, in Recurrent Medulloblastoma and Ependymoma	2	62	Drug: Irinotecan Drug: Temozolomide Drug: Bevacizumab Drug: Omburtamab I-131 Drug: Liothyronine Drug: SSKI Drug: Dexamethasone Drug: Antipyretic Drug: Antihistamine Drug: anti-emetics

## Data Availability

Not applicable.

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
