# Peer review of "Angiogenesis and Lymphangiogenesis in Medulloblastoma Development"

_biology, 2023, doi:10.3390/biology12071028_

Round 1
Reviewer 1 Report
Comments and Suggestions for Authors:
Overall, the review “Angiogenesis and lymphangiogenesis in medulloblastoma development” by Penco-Campillo et al is well written and well organized. The English used is excellent, thus there are no corrections or comments to make on the language used. Images and tables are graphically well made and quite sharp.
The manuscript manages to embrace the topic of angiogenesis in medulloblastoma in a very comprehensive manner, ranging from the role it has in tumor dissemination and growth to its role in resistance to therapies. The molecular complexity at the basis of this process has also been reported in depth.
The very few comments about this review are:
· -In the legend of Figure 1, you should expand the abbreviation used in the image. This aspect should be checked in the legends of all the figures through the text.
· - Line 96: there is an extra comma between the word "the" and "median".
· - The formatting in some places in the text is not uniform. In paragraph 4 the text should be justified. In sub-paragraph 4.3.3. the paragraph is not divided by spaces. In paragraph 5 the spaces are shorter than in the rest of the manuscript. This type of check should be done throughout the text.
· - While the images are well done and quite descriptive, it may be more effective for readers to use brighter colors in some of them.
· - If possible and within the scope of the topic, you should add a paragraph that focuses on novel anti-angiogenic therapeutic approaches, such as the possible use of miRNAs, extracellular vesicles, and nanotechnologies.
· -In the conclusions section, you should better summarize the possible future directions on anti-angiogenic therapies in MB.
Reviewer 2 Report
The authors have presented a comprehensive review manuscript spanning 35 pages titled "Angiogenesis and Lymphangiogenesis in Medulloblastoma Development." This manuscript is intended to contribute to a special issue focused on "Cellular and Molecular Mechanisms of Medulloblastoma and Its Therapeutic Targets." The authors have made an effort to integrate modern medulloblastoma diagnostics, ranging from histopathology to molecular diagnosis, with recent findings on lymphatic-like structures primarily found in the meninges. However, the reviewer could not find convincing references that significantly link these aspects to medulloblastoma. While this may be part of a new research project, it does not seem to fit well within a review, unless discussed with limitations in mind.
Despite the authors' considerable effort in summarizing recent and evolving aspects of medulloblastoma research and management, the manuscript requires a major revision to further improve its quality.
Major and minor issues:
1. L.1 + L. 24: "second most frequent" – Isn’t it actually the most frequent? Please provide a reference to support this claim.
2. L. 29: Medulloblastoma is no longer classified as a PNET (primitive neuroectodermal tumor).
3. L. 138-156: This paragraph should be rephrased and expanded upon. Mentioning recently discovered lymphatic-like structures in the meninges and their potential relation to medulloblastoma is highly speculative, especially in the context of a review. Additionally, references to medulloblastoma are lacking, while an important citation that should be discussed is Garzia et al. 2005: "A Hematogenous Route for Medulloblastoma Leptomeningeal Metastases."
4. L. 174-531 (Chapter 4): Although this section covers important and general aspects of lymphatics, it lacks a specific connection to medulloblastoma. It is suggested to either remove or shorten this section. Alternatively, these aspects could be used in a new manuscript focusing on tumors located outside the brain.
5. L. 534-775 (Chapter 5): This section does not contain significant references or direct relevance to medulloblastoma. It is recommended to concentrate on citing publications specifically related to medulloblastoma in this section.
Reviewer 3 Report
Dear Authors,
I congratulate for your work depicting the importance of angiogenesis and lymphomagenesis in MB tumours.
I believe you did an excellent work and provided outstanding information also in the form of beautiful figures.
To this reviewer, your review paper can be published in Biology journal as it is.
With my best wishes for your future work.
Round 2
Reviewer 2 Report
The manuscript improved significantly. Although the reviewer is not convinced at all with the author's views on the suggested importance of lymphatics in the head and any role of medulloblastoma metastasis, the reviewer supports the publication, since it clearly reflects the authors opinion and really may induce scientific discussions leading to further clarification in this field.